# Sex-Specific Differences in Cardiovascular Risk, Risk Factors and Risk Management in the Peripheral Arterial Disease Population

**DOI:** 10.3390/diagnostics12040808

**Published:** 2022-03-25

**Authors:** Anna Louise Pouncey, Mark Woodward

**Affiliations:** 1Department of Vascular Surgery, Division of Surgery and Cancer, Faculty of Medicine, Imperial College London, QEQM, St Mary’s Hospital, Praed Street, London W2 1NY, UK; 2The George Institute for Global Health, School of Public Health, Imperial College London, London W12 0BZ, UK; markw@georgeinstitute.org.au; 3The George Institute for Global Health, University of New South Wales, Sydney, NSW 2050, Australia

**Keywords:** peripheral arterial disease, cardiovascular risk, sex

## Abstract

Cardiovascular disease (CVD) is the leading cause of mortality in women worldwide but has been primarily recognised as a man’s disease. The major components of CVD are ischaemic heart disease (IHD), stroke and peripheral arterial disease (PAD). Compared with IHD or stroke, individuals with PAD are at significantly greater risk of major cardiovascular events. Despite this, they are less likely to receive preventative treatment than those with IHD. Women are at least as affected by PAD as men, but major sex-specific knowledge gaps exist in the understanding of relevant CVD risk factors and efficacy of treatment. This prompted the American Heart Association to issue a “call to action” for PAD in women, in 2012. Despite this, PAD and CVD risk in women continues to be under-recognised, leading to a loss of opportunity to moderate and prevent CVD morbidity. This review outlines current evidence regarding cardiovascular risk in women and men with PAD, the relative significance of traditional and non-traditional risk factors and sex differences in cardiovascular risk management.

## 1. Introduction

Cardiovascular disease (CVD) kills women [1]. In fact, it is the leading cause of mortality worldwide, outnumbering conventional women’s health targets (childbirth, gynaecological and breast cancer) combined [2]. As men have a higher incidence at all ages, CVD has previously been primarily recognised, investigated, and treated as a man’s disease. However, in reality there is minimal difference in lifetime risk (67% vs. 66% remaining lifetime risk at 55 years, *p* = non-significant (ns)) due to the increased life expectancy of women [3]. The treatment of women as a minority sub-population has led to a lack of evidence-based treatment, reduced personal and clinician awareness of CVD risk in women, and a lower proportion of women receiving adequate care [4,5,6,7]. Initiatives such as the American Heart Association’s “Go Red for Women” are making efforts to counter this, promoting dedicated research, patient and clinician awareness and a higher standard of care for women with cardiac disease [5].

The major components of CVD are ischaemic heart disease (IHD), stroke and peripheral arterial disease (PAD). In 2012, the American Heart Association issued a call to action for PAD in women and promoted the translation of “Go Red for Women” to the field of vascular surgery. It highlighted that women are at least as affected by PAD as men, but that major sex-specific knowledge gaps exist in the understanding of relevant CVD risk factors, sex-specific clinical presentation and the efficacy of diagnostic tests and treatment pathways in PAD [8]. Despite this initiative, PAD remains under-recognised as a significant cause of CVD morbidity and mortality in women, which may lead to a delayed diagnosis and insufficient moderation of risk [4]. This review aims to outline current evidence regarding cardiovascular risk in women and men with PAD, the relative significance of traditional and non-traditional risk factors and sex differences in cardiovascular risk management.

### 1.1. Peripheral Arterial Disease and Cardiovascular Risk—Cardiology’s Poor Cousin

Peripheral arterial disease (PAD) comprises atherosclerotic disease of one or more peripheral arteries, and carries similar morbidity and mortality risks, and comparable healthcare costs, to ischaemic heart disease [9,10]. In the UK, symptomatic PAD occurs in ~4.5% of the population (~3% aged ≥50 years and 7%~aged ≥70 years) [11,12], and overall PAD affects ≥200 million people worldwide [13]. In addition to a restriction in mobility, a reduction in quality of life, ulceration and amputation, individuals with a diagnosis of PAD also have 2.5 times the risk of incident myocardial infarction (MI) and 3.1 times the risk of incident stroke, compared with those without PAD [14,15]. The REACH registry (Reduction of Atherothrombosis for Continued Health) demonstrated that, even when compared with individuals with ischaemic stroke or MI, individuals with PAD are at significantly greater risk of major cardiovascular events, hospitalisation and intervention; 52% of the PAD population have concomitant IHD and 23% cerebrovascular disease (Appendix A) [16]. Despite this, cardiovascular risk in the PAD population is under-recognised by clinicians and the public. As a result, those with PAD are significantly less likely to receive secondary preventative treatment than those following an MI [17,18].

### 1.2. Women with Peripheral Vascular Disease—A Greater Cardiovascular Risk?

Women have a similar overall prevalence of PAD, but a higher prevalence of asymptomatic or atypical PAD, and in the USA, amongst those aged 70 and above, exhibit a greater PAD burden than men [8,14]. Use of the ankle–brachial pressure index (ABPI) identifies 3–5 times the number of PAD cases in women than those diagnosed on clinical history alone, and women with PAD are at 2–4 times the risk of cardiovascular events and death compared with women without PAD [19]. Overall, a similar association between ABPI values, mortality, and major coronary events has been demonstrated for both sexes, with women at comparably greater risk than men at lower and higher ABPI’s (<0.7 & >1.4) [8]. In a Canadian population-based cohort study it was also observed that women with PAD were more likely to suffer acute MI than men (Hazard Ratio (HR) 1.15, 95% Confidence Interval (CI) 1.00–1.31) [20]. This may reflect later diagnosis of PAD in women, resulting in loss of opportunity for secondary cardiovascular prevention in the latent window of the disease.

Women are noted to have lower functional performance and greater exertional leg pain than men, which may be associated with reduction in muscle strength or concomitant morbidity, such as spinal stenosis [21,22]. These differences are reflected in lower quality of life scores for physical function and general health [23]. In general, women present around 10 years later than men, more often with critical limb ischaemia, and complex multi-level or femoropopliteal disease [24,25,26], and are less likely to be offered an intervention [27].

Reporting of outcomes for women with PAD are inconsistent. In the USA, it has been reported that women are at a higher risk of bleeding, infection and mortality following vascular procedures, but in Sweden, following adjustment for age, women were not at greater risk of poor outcomes [28,29]. A meta-analysis, of 40 studies, examining outcomes after lower extremity revascularisation, demonstrated that women have inferior short-term outcomes, with an increased risk of 30-day mortality (Odds Ratio (OR) 1.31, 95% CI 1.11–1.55), stroke (OR 1.35, 95% CI 1.19–1.53), cardiac event (OR 1.21 95% CI 1.16–1.26), and early graft thrombosis (OR 1.56, 95% CI 1.29–1.90) for both open and endovascular procedures [30]. No significant difference in the risk of short-term reintervention (OR 1.06, 95% CI 0.73–1.54) and long-term patency were observed. Following endovascular revascularisation long-term survival was similar for both sexes, but was inferior for women following open revascularisation (HR 1.21, 95% CI 1.01–1.44) [30], while open bypass surgery seems to carry a greater risk of graft thrombosis (OR 1.29, *p* = 0.005), limb loss and mortality for women [27,31,32], a higher rate of technical success (91.2% vs. 89.1%, *p* = 0.014) and better amputation free survival, have been reported for women undergoing endovascular intervention, despite increased embolic events, major bleeding and an inferior functional outcome, with more frequent discharge to a nursing home [25,26,33].

### 1.3. Traditional Cardiovascular Risk Factors

Traditional risk factors for CVD (hypertension, hyperlipidaemia, diabetes mellitus, smoking and obesity) are present in the majority of individuals with PAD, shared between the sexes, and form the cornerstone of cardiovascular risk stratification and secondary preventative treatment [34,35]. However, the timing and nature of traditional risk factor effects on PAD and CVD outcomes may differ for women and men.

Hypertension is associated with double the risk of PAD for women compared with men [36]. Blood pressure control is a powerful modifiable risk factor; a reduction of 10 mmHg in systolic pressure decreases the risk of stroke-related mortality by 40% and cardiac mortality by 30% [37,38]. In general, blood pressure targets are applied equally to both sexes; however, a study of over 27,000 participants from the Framingham Heart Study, Multi-Ethnic Study of Atherosclerosis, Atherosclerosis Risk in Communities Study and Coronary Artery Risk Development in Young Adults Study, it was noted that the risk of CVD proportionately increased at a lower range of systolic blood pressure in women compared with men. Indeed, the risk of myocardial infarction for women with an SBP of 110–119 was equivalent to the risk for men with an SBP of >160 (HR 1.64, 95% CI 1.20–2.25, vs. HR 1.62, 95% CI 1.14–2.30) [39]. Although not specific to PAD, this suggests that there may be unrecognised sex-specific differences in optimal blood pressure targets.

Individuals with a diagnosis of diabetes are more likely to develop PAD and those with an ABPI <0.9 have a 67% increased risk of cardiac death [40]. While the risk of lower extremity amputation, amongst those with diabetes, is observed to be higher for men (pooled adjusted OR 1.44, 95% CI 1.24–1.67), women with insulin-dependent diabetes mellitus have a 40% excess risk of death, and an 86% excess risk of cardiovascular mortality [41,42,43]. Diabetes mellitus is also associated with an additional 44% excess risk of coronary artery disease for women (above the 2-fold increase for men) as well as a 27% excess risk of stroke [44]. These sex-specific differences may arise due differences in hormonal signalling and adipose deposition, meaning that greater metabolic deterioration is needed for women to develop diabetes. Indeed, at the time of diagnosis, a greater increase in cardiovascular risk factors is observed in women relative to men [45]. Obesity and resultant metabolic aberration and endothelial dysfunction may also supersede the cardiovascular protective effects of female sex hormones [45]. In addition, while there is little evidence to suggest that women with diabetes receive a different standard of care to men, women with diabetes are less likely to achieve adequate blood pressure and lipid control [46,47]. It has also been reported that women with diabetes, despite similar adherence, do not respond as well to exercise rehabilitation for PAD [48].

Chronic kidney disease (CKD) is associated with an increased risk of PAD for both men and women. In the National Health and Nutrition Examination Survey, 24% of the population with CKD (stage 3 or above) were demonstrated to have an ABPI <0.9, compared with 3.7% of those without (*p* < 0.001) [49]. However, the nature of the risk appears to be different for the sexes. The Chronic Renal Insufficiency cohort, a multicentre prospective study of 3,174 patients aged 21–74 with renal disease, demonstrated that women are at increased risk for PAD at younger ages, with a subdistribution HR (SHR) of 2.57 (95% CI 1.27–5.20) amongst those younger than 40 years of age. However, while the risk of PAD gradually increased with age for men, the same was not observed for women, resulting in a similar risk difference in those aged 70 years and above (SHR 1.05, 95% CI 0.66–1.67) [50]. This finding in the CKD population is contrary to the later onset of PAD for women that is normally observed, and further work is needed to elucidate how sex-specific biological or clinical differences may contribute to the development of PAD in this population.

Smoking is associated with 2.3 times the risk of development of symptomatic PAD, and presents a significant modifiable risk factor [51]. Smoking is much more prevalent amongst men worldwide, but varies over time and with geographic location [51]. It has been reported that for the same smoking history, women have a 25% higher risk of IHD compared with men, which is postulated to be secondary to synergistic factors, such as the use of combined oral contraception [51]. Indeed, in a meta-analysis, a multiple-adjusted pooled relative risk ratio (female to male) of smoking to non-smoking for coronary heart disease was found to be 1.25 [52]. In addition, women have 31% lower odds of successful smoking cessation, and are thought to respond differently to smoking cessation pathways [53].

### 1.4. Non-Traditional Cardiovascular Risk Factors

The Multi-Ethnic Study of Atherosclerosis identified that PAD, as defined by a low (<1.0) or high (>1.3) ABPI, was still common in a cohort of 1932 participants without traditional risk factors, affecting 9% and 7.8% of participants, respectively, and was also associated with concomitant IHD [54]. This suggests a significant knowledge gap in our understanding of relevant risk factors for CVD and PAD, which is likely to be more pronounced for women. Indeed, in the Heart and Soul study, a prospective study of 1024 participants with IHD, while traditional risk factors were significant predictors for the development of PAD in men, depression was observed to have greater importance in women [55]. This implies that hitherto un-appreciated, or “non-traditional”, risk factors may have greater significance in women and carry the potential to identify and treat women prior to CVD morbidity in the latent stages of disease.

### 1.5. Sex-Specific Risk Factors

In addition to traditional risk factors, it is increasingly recognised that factors unique to women also contribute to CVD (See Appendix A) [5]. This had led to a revision of risk-stratification and screening guidance for cardiac disease in women [56].

Adverse pregnancy outcomes are associated with increased CVD risk and affect 3–20% of pregnancies [56]. Preterm delivery is associated with increased risk of CVD, and women are noted to have subclinical atherosclerosis at ~10 years post-delivery [57]. Around 25% of this risk is attributable to the later development of hypertension, diabetes and hyperlipidaemia, but additional mechanisms are not yet elucidated [56]. Hypertension in pregnancy is an independent risk factor for PAD, adjusting for age, smoking, hypertension and diabetes [58]. Furthermore, placentally mediated conditions (pre-eclampsia, gestation hypertension, placental abruption or infarction) double the risk of IHD in the ensuing decade and quadruple the risk when associated with intrauterine foetal death [59]. Gestational diabetes is associated with an increased risk of CVD, even amongst those who do not subsequently develop diabetes (relative risk (RR) 1.56, 95% CI 1.04–2.32) [1,57], and pregnancy loss, polycystic ovarian syndrome (PCOS) and premature menarche are also associated with increased CVD risk [56]. Conversely, breastfeeding for >4 months has been shown to be associated with a 30% reduction in the risk of hypertension and a 20% reduction in CVD. This may be secondary to the acceleration of metabolic recovery following pregnancy, but is observed to be independent of BMI [57]. Awareness of these conditions, which can efficiently be identified at an early time point when many women access healthcare, could enable monitoring and appropriate lifestyle modifications, and ultimately prevent CVD in later life.

The use of hormonal contraceptives has been shown to impair macrovascular endothelial function, dependent on the progestin type and route of administration [60]. An association between the use of oestrogen-based or androgenic contraceptives may also induce dyslipidaemia, increasing CVD risk in the long term [56]. A case–control study in the Netherlands reported an increased risk of PAD associated with hormonal contraceptive use amongst women 18–49 years of age (adjusted OR 3.8, 95% CI 2.4–5.9) [61]. However, a cross-sectional analysis of 887 women in the KORA-F4 study found no significant association with oral contraceptives, but observed a reduction in PAD risk with late menarche (onset >15 years, OR 0.48, 95% CI 0.24–0.98) [61]. PCOS is associated with hormonal dysregulation, excess androgens, ovarian dysfunction and an adverse cardiovascular risk profile with increased risk of central adiposity, hypertension, dyslipidaemia and insulin resistance [56]. Although the extent of the hormonal effect, compared with the effect of associated cardiometabolic comorbidity, is not determined, a diagnosis of PCOS is associated with premature carotid atherosclerosis, and an increased risk of MI (OR 2.57, 95% CI 1.37–4.82) and stroke (OR 1.96, 95% CI 1.56–2.47) [62,63,64]. Oestrogen is generally postulated to have protective effects prior to the menopause, with degeneration in endothelial function observed a decade later in women than men, and an early menopause is associated with increased CVD risk [1,65,66]. However, these effects may be over-simplified or over-stated. A study examining the risk–benefit of unopposed oestrogen hormone replacement therapy after the menopause has suggested no cardiovascular benefit, but rather an increased risk of PAD (HR 1.63, 95% CI 1.05–2.51) [67].

### 1.6. Sex-Predominant Risk Factors

Psychological morbidity is an important, modifiable, and often overlooked risk factor for CVD, and is associated with an increased risk of PAD [55,68]. A single-centre study in America demonstrated that, following PAD revascularisation procedures, patients with depression were at greater risk of death or major adverse cardiovascular events (adjusted HR 2.05, 95% CI 1.16–2.86), and progression of contralateral disease (HR 2.20, 95% CI 1.22–3.96) [21]. Associations between depression and death following coronary artery bypass surgery have been demonstrated, and the presence of mental-stress-induced ischaemia amongst the stable CAD population is associated with an increased risk of cardiovascular death or MI (HR 2.0, 95% CI 1.1–3.7) [69,70,71]. This increased CVD risk may arise secondary to changes in microvascular tone, blood pressure, endothelial dysfunction and heightened platelet aggregation, which have been observed in depression and anxiety [68,71,72,73,74]. In a study of 444 patients in Dutch vascular outpatient clinics, approximately 40% of women aged <65 years, with a diagnosis of PAD of 6 months duration, demonstrated significant depressive symptoms. Adjusting for PAD severity, demographics, and clinical factors, this equates to four times the rate of depression in men aged >65 years [75]. Depression was the strongest independent risk factor for PAD amongst women in the Heart and Soul Study (OR 3.26, 95% CI 1.09–9.77) and therefore clearly warrants recognition and dedicated investigation [55].

Autoimmune diseases, such as systemic lupus erythematosus or rheumatoid arthritis, are more predominant in women than men (8.4% vs. 5.1%, *p* < 0.001), and share inflammatory pathways with atherosclerosis [5,76,77]. Rheumatoid arthritis is a risk factor for PAD (HR 2.29, 95% CI 1.20–4.34) and is associated with an increased risk of myocardial infarction, aneurysmal disease and CVD mortality [5,76,78,79,80]. Non-traditional risk factors, such as inflammation, low BMI, sarcopenia, and rheumatoid arthritis-specific risk factors (i.e., rheumatoid factor positivity and markers of disease severity), appear to be of greater importance for CVD risk in women, compared with men who demonstrate more traditional risk factors [51]. The use of pharmacological intervention can be complex in rheumatological disease. Anti-inflammatory medication, such as steroids, may convey a cardioprotective effect, but can also aggravate hypertension, dyslipidaemia, and insulin resistance. The monitoring of lipid-lowering agents, such as statins, may also prove difficult during acute inflammation [76]. Further inflammatory autoimmune disease, such as systemic sclerosis, antiphospholipid syndrome, polymyalgia rheumatica and giant cell arteritis, are also more predominant in women, but limited data on CVD risk are reported, although giant cell arteritis is also associated with an increased risk of PAD and CVD events [51].

Individuals with PAD have an increased risk of concomitant cancer compared with the general population, likely secondary to common risk factors [45]. Breast cancer and its treatments are a significant sex-predominant factor. Double the risk of CVD mortality is observed at 7 years following treatment, with the greatest risk observed for pre-menopausal women who had received chemotherapy [5,81]. The long-term use of aromatase inhibitors is also associated with impaired endothelial function and an increased risk of cardiovascular mortality (HR 1.5, 95% CI 1.11–2.04) and heart failure [82]. In addition to the increased risk of thrombotic events associated with cancer itself, chemotherapeutic agents can cause endothelial dysfunction and cytotoxicity, the suppression of anti-inflammatory and reparative functions, platelet activation and a reduction in anticoagulant activity, and as such careful consideration of the risk–benefit and pre-disposing cardiovascular risk must be considered prior to treatment. One such example is vascular endothelial growth factor inhibitors, which can suppress angiogenesis (vital for the production of collateral pathways in PAD), increase hypertension and roughly double the risk of arterial thrombosis [51]. Significant PAD risk is also observed with tyrosine kinase inhibitors for haematological malignancy, most notably with the use of Nilotinib, which carries a 15% risk of an acute arterial event at 2 years and a 33% risk of PAD at 10 years [51].

### 1.7. Intersectionality: Gender, Race and Sociodemographic Factors

Gender, as opposed to the biological effect of sex, is a social construct, which can be subject to external influence. An intersectional approach to gender examines the effect of a dynamic interaction between gender and complex social categories, such as race and social deprivation, on the individual. Each factor may conversely present an advantage or disadvantage, and can convey CVD risk through effects on health behaviours such as physical activity levels, alcohol consumption, smoking and differences in psychosocial stressors and environmental exposures, such as airborne pollution [66,83]. The effects of interaction between gender and social categorisations on PAD risk remain to be clarified, but are likely to be cumulative [83]. Indeed, the highest prevalence of PAD in the USA is amongst black women, while women with PAD have also been observed to be of poorer socioeconomic status [84]. In the PREVENT III (Project of Ex Vivo Vein Graft Engineering via Transfection III) trial, all black participants were at increased risk of graft failure and amputation, but black women were at greatest risk (HR 2.38, 95% CI 1.18–4.83, for amputation) [85]. The INTERHEART study also found that modifiable risk factors for cardiac disease were significantly higher among women than men (74.3%, 95% CI 67.9–80.7, vs. 67.3%, 95% CI 63.9–70.8) and that psychosocial risk was more significant [81]. Therefore, efforts to address intersectionality, and its effect on PAD and CVD risk, are warranted and may help to resolve the differences observed.

### 1.8. Sex-Specific Differences in Traditional Cardiovascular Risk Management 

Women with PAD are less likely to receive optimum medical care (defined as an antiplatelet, statin, angiotensin-converting enzyme inhibitor and smoking cessation) than men (29% vs. 54%, *p* < 0.038), but both sexes benefit significantly when they do, as evidenced by the reduction in major adverse cardiovascular events [86,87].

High-intensity statin therapy for PAD is associated with a 26% risk reduction in all-cause mortality and a 33% reduction in amputation [88]. The reporting of sex-specific analyses in statin-efficacy trials are inadequate but there is evidence to suggest benefit for both sexes [89]. Data from the Vascular Quality Initiative between 2010 and 2013 report that women with PAD were less likely to receive statin therapy (64% vs. 70%, *p* < 0.001) [87]. This may be due to a lack of recognition, discontinuation, or refusal, and a greater prevalence of statin intolerance has been reported amongst women [18]. Indeed, a study in the USA found that women who met the criteria of the American Heart Association Cholesterol Guideline to be recommended statin therapy were also less likely to receive treatment (67% vs. 78.4%, *p* < 0.001) or to receive the recommended dose (36.7% vs. 45.2%, *p* < 0.001). Overall, more women cited a lack of prescription (18.6% vs. 13.5%, *p* < 0.001), and a higher rate of discontinuation (10.9% vs. 6.1%, *p* < 0.001) or refusal (3.6% vs. 2.0%, *p* < 0.001) [90]. A higher rate of the premature discontinuation of drug therapy amongst women has also been reported in major cardiovascular outcome trials [91].

The prescription of antiplatelet therapy with PAD is associated with a 23% reduction in major vascular events [92]. Data from the Vascular Quality Initiative between 2010 and 2013 report that women with PAD were only marginally less likely to receive antiplatelet (78% vs. 80%, *p* < 0.001) and anticoagulant therapy (8.4% vs. 9.3%, *p* = 0.017), compared with men with PAD [87]. In addition, sex-specific differences in platelet function are reported, with women exhibiting a higher platelet reactivity [93]. However, it is unclear whether this represents a true biological difference, or has relevance for the efficacy of anti-platelet therapy [94,95]. A meta-analysis of trials comparing antithrombotic therapy in PAD found no effect of sex on graft occlusion or cardiovascular events with the use of anticoagulants, but results were limited by under-enrolment, with a participation/prevalence ratio of 0.41 [96].

## 2. Discussion

It has been a decade since the call to action by the American Heart Association for women and peripheral arterial disease, but can we honestly say there has been appreciable change [8]? Even if we could assure ourselves that the women selected for treatment have similar technical outcomes to men, we know that they suffer. They experience greater pain than men, have reduced quality of life, and are at greater risk of concurrent cardiovascular events [8,21,22,23].

With later PAD diagnosis we are missing an opportunity to reduce CVD risk during the latent period of disease. Attention should be paid to increasing our understanding of clinical presentation in women, as well as the relevant risk factors, which may enable screening and preventative treatment in populations at risk [19]. Although the recognition and control of traditional risk factors are important for both sexes, they appear to have a differential effect and may have lesser overall significance for women with PAD than men [46,49]. Nonetheless, the treatment of traditional risk factors conveys a protective effect for both sexes and efforts should be made to ensure the equitable provision of optimal care for women and men [73]. The recognition of non-traditional sex-specific (e.g., pregnancy-related) or sex-predominant (e.g., depression) factors may enable appropriate risk stratification and prevention [52,57]. This requires further investigation and is likely to require the implementation of designated treatment pathways and strategies [5].

To achieve appreciable change, clinicians and researchers will need to look beyond the boundaries of each cardiovascular specialty [48]. Similarly, sex as a biological construct cannot be taken in isolation, but should be explored and reported in the context of intersectionality, with consideration of factors including gender, race, socioeconomic status and disability [83]. This requires dedicated research but also can commence with adequate research enrolment and reporting [84].

Despite women making up roughly 50% of the world’s population, and even in the face of some journals and funding bodies requiring women to be well represented in research, the reporting of sex-disaggregated data is by no means regular and women continue to be under-represented in cardiovascular randomised controlled trials [18,85]. This not only inhibits research to identify sex differences for the benefit of both sexes, such as the current article, but also contributes to inadequate care for women, and ultimately costs disability-adjusted life years. Missed opportunities can only be identified if robust evidence is available, and the current picture in PAD is poor where sex disparities are concerned. We thus plea that future researchers in this field routinely include a fair percentage of both sexes, unless there is good reason to do otherwise, and report sex-disaggregated data as an a priori subgroup analysis—at least in a web appendix should there be no evidence of a difference, as this is important information for subsequent systematic reviews and meta-analyses. Perhaps then, another decade on from the timely call to action from the American Heart Association, we will be able to write a very different review in the year 2032.

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
