# Peer review of "Sex-Specific Differences in Cardiovascular Risk, Risk Factors and Risk Management in the Peripheral Arterial Disease Population"

_diagnostics, 2022, doi:10.3390/diagnostics12040808_

Round 1

Reviewer 1 Report

Sex-specific data focused on cardiovascular disease are increasing steadily, but it is not routinely collected, applied, or included in clinical guidelines.

In addition, there is a problem of underestimation by clinicians of patients with peripheral arterial disease (PAD), the complications of which are comparable to those of coronary artery disease, and the risk of myocardial infarction in patients with PAD is higher. It is known that secondary prevention is more often prescribed for patients with myocardial infarction, coronary heart disease and stroke, but not for patients with peripheral arterial disease. Prevention and treatment during the latent course of the disease could reduce the risk of cardiovascular disease in patients with PAD, but this opportunity is often missed.

Men and women are susceptible to developing PAD, but women often present with asymptomatic or atypical disease and so they seek medical attention in the later stages of the disease. A number of studies show that measuring the ankle-brachial pressure index (ABPI) in women reveals a high number of PAD in those who have not previously presented to the hospital.

Risk factors for PAD are similar to coronary artery disease and other atherosclerotic diseases. The article covers quite well paragraphs 1.1-1.7, in which these factors are described in relation to women.

The influence of purely female factors (the influence of hormones, pregnancy, childbirth, taking contraceptives, etc.) on the risk of developing cardiovascular diseases is well described. Besides, the relevance of this article is emphasized by the growing number of reports in which the emphasis is on considering the possibility of including reproductive risk factors as part of the assessment of the risk of developing cardiovascular diseases in clinical guidelines.

In my opinion, information on the occurrence of chronic kidney disease (CKD) among women can be added to paragraph 1.3. It is known that CKD is known risk factor for accelerated atherosclerotic process. For example, a study by the Wang group reports that Women <70 years of age with chronic kidney disease have 1.5 times higher risk of developing PAD than men.

In paragraph 1.4 where the authors describe information about taking oral contraceptives, they can focus on the fact that women taking any generation oral contraceptives have higher odds of having PAD as compared to women taking no oral contraceptives. It would also be nice to add the KORA F4 study here. This study mentions the association of reproductive factors with peripheral arterial disease in women.

In paragraph 1.5 the authors can add information about the association of mental stress-induced myocardial ischemia with cardiovascular events. The influence of acute psychological stress on cardiovascular disease is studying now. Acute psychological stress may induce arteriolar vasoconstriction and reduce blood flow to the vital organs, while simultaneously increasing blood pressure, which contributes to the development of cardiovascular disease.

My suggestions do not detract from the positive impression of the article as a whole.

Author Response

In my opinion, information on the occurrence of chronic kidney disease (CKD) among women can be added to paragraph 1.3. It is known that CKD is known risk factor for accelerated atherosclerotic process. For example, a study by the Wang group reports that Women <70 years of age with chronic kidney disease have 1.5 times higher risk of developing PAD than men.

Many thanks for this helpful suggestion, we have added the following to the text (lines 143-169):

“Chronic kidney disease (CKD) is associated with an increased risk of PAD for both men and women. In the National Health and Nutrition Examination Survey, 24% of the population with CKD (stage 3 or above) were demonstrated to have an ABPI <0.9, compared to 3.7% of those without (p<0.001)[49]. However, the nature of the risk appears to be different for the sexes. The Chronic Renal Insufficiency cohort, a multicentre prospective study of 3,174 patients aged 21-74 with renal disease, demonstrated that women are at increased risk for PAD at younger ages, with a subdistribution HR [SHR} of 2.57 (95% CI 1.27-5.20) amongst those younger than 40 years of age. However, while the risk of PAD gradually increased with age for men, the same was not observed for women, resulting in a similar risk difference in those aged 70 years and above (SHR 1.05, 95% CI 0.66-1.67)[50]. This finding in the CKD population is contrary to the later onset of PAD for women, which is normally observed, and further work is needed to elucidate how sex-specific biological or clinical differences may contribute to the development of PAD in this population.”

In paragraph 1.4 where the authors describe information about taking oral contraceptives, they can focus on the fact that women taking any generation oral contraceptives have higher odds of having PAD as compared to women taking no oral contraceptives. It would also be nice to add the KORA F4 study here. This study mentions the association of reproductive factors with peripheral arterial disease in women.

Many thanks for this helpful suggestion, we have added the following to the text (lines 214-223):

“Use of hormonal contraceptives have been shown to impair macrovascular endothelial function, dependent on the progestin type and route of administration[60]. An association between use of oestrogen-based or androgenic contraceptives may also induce dyslipidaemia, increasing CVD risk in the long term[56]. A case-control study in the Netherlands reported an increased risk of PAD associated with hormonal contraceptive use amongst women, 18-49 years of age (adjusted OR 3.8, 95% CI 2.4-5.9)[61]. However, a cross-sectional analysis of 887 women in the KORA-F4 study, found no significant association with oral contraceptives, but observed a reduction in PAD risk with late menarche, (onset >15 years, OR 0.48, 95% CI 0.24-0.98)[61].”

In paragraph 1.5 the authors can add information about the association of mental stress-induced myocardial ischemia with cardiovascular events. The influence of acute psychological stress on cardiovascular disease is studying now. Acute psychological stress may induce arteriolar vasoconstriction and reduce blood flow to the vital organs, while simultaneously increasing blood pressure, which contributes to the development of cardiovascular disease.”

Many thanks for this helpful suggestion, we have added the following to the text (lines 242-248):

“Associations between depression and death following coronary artery bypass surgery have been demonstrated, and the presence of mental stress-induced ischaemia amongst the stable CAD population is associated with an increased risk of cardiovascular death or MI (HR 2.0, 95% CI 1.1-3.7[63–65] This increased CVD risk may arise secondary to changes in microvascular tone, blood pressure, endothelial dysfunction and heightened platelet aggregation, which have been observed in depression and anxiety[62,65–68]."

Reviewer 2 Report

I was invited to revise the paper entitled "Sex-specific Differences in Cardiovascular Risk, Risk Factors and Risk Management in the Peripheral Arterial Disease Population". It was a review that aimed to summarize all evidence about gender differences in rask factors associated to PAD.

The paper is well structured and examined the great part of factors associated to PAD. The paper was well structured e easy to read. I have some observations:

  • Authors did not discussed the impact of diabetes on PAD. Many recent papers focused on gender differences in lower extremeties amputations, such as 10.1177/1534734614545872, 10.1016/j.fas.2020.01.005 and 10.1016/j.jvs.2016.11.030.
  • No mention was made about outcomes after revascularization;
  • In my opinion, a focus on drug therapy and PCOS should be made.

Minor observations:

  • Figure S1 should be presented in main text;
  • In my opinion this paper does not proerly match the aim of this journal.

Author Response

Authors did not discussed the impact of diabetes on PAD. Many recent papers focused on gender differences in lower extremeties amputations, such as 10.1177/1534734614545872, 10.1016/j.fas.2020.01.005 and 10.1016/j.jvs.2016.11.030. 

Many thanks for this helpful suggestion, we have added the following to the text (lines 126-131):

RE: 10.1177/1534734614545872, 10.1016/j.fas.2020.01.005 

“Individuals with a diagnosis of diabetes are more likely to develop PAD and those with an ABPI <0.9 have a 67% increased risk of cardiac death[38]. While the risk of lower extremity amputation, amongst those with diabetes, is observed to be higher for men (pooled adjusted OR 1.44, 95% CI 1.24-1.67), women with insulin-dependent diabetes mellitus have a 40% excess risk of death, and 86% excess risk of cardiovascular mortality[39–41].”

RE: 10.1016/j.jvs.2016.11.030. – included in text within response to comment below 

No mention was made about outcomes after revascularization;

Many thanks for this helpful suggestion, we have added the following to the text (lines 86-107):

“Reporting of outcomes for women with PAD are inconsistent. In the USA, it has been reported that women are at a higher risk of bleeding, infection and mortality following vascular procedures, but in Sweden, following adjustment for age, women were not at greater risk of poor outcomes[28,29]. A meta-analysis, of 40 studies, examining outcomes after lower extremity revascularization, demonstrated that women have inferior short-term outcomes, with an increased risk of 30-day mortality (Odds Ratio [OR] 1.31, 95% CI 1.11-1.55), stroke (OR 1.35, 95% CI 1.19-1.53), cardiac event (OR 1.21 95% CI 1.16-1.26), and early graft thrombosis (OR 1.56, 95% CI 1.29-1.90) for both open and endovascular procedures[30]. No significant difference in the risk of short-term reintervention (OR 1.06, 95% CI 0.73-1.54) and long-term patency were observed. Following endovascular revascularization long-term survival was similar for both sexes, but was inferior for women following open revascularization (HR 1.21, 95% CI 1.01-1.44)[30]. While, open bypass surgery seems to carry a greater risk of graft thrombosis (OR 1.29, p=0.005), limb loss and mortality for women[27,31,32], a higher rate of technical success (91.2% vs. 89.1%, p = 0.014) and better amputation free survival, have been reported for women undergoing endovascular intervention, despite increased embolic events, major bleeding and an inferior functional outcome, with more frequent discharge to a nursing home[25,26,33].”

In my opinion, a focus on drug therapy and PCOS should be made.

Many thanks for this helpful suggestion, we have added the following to the text (lines 214-235):

“Use of hormonal contraceptives have been shown to impair macrovascular endothelial function, dependent on the progestin type and route of administration[60]. An association between use of oestrogen-based or androgenic contraceptives may also induce dyslipidaemia, increasing CVD risk in the long term[56]. A case-control study in the Netherlands reported an increased risk of PAD associated with hormonal contraceptive use amongst women, 18-49 years of age (adjusted OR 3.8, 95% CI 2.4-5.9)[61]. However, a cross-sectional analysis of 887 women in the KORA-F4 study, found no significant association with oral contraceptives, but observed a reduction in PAD risk with late menarche, (onset >15 years, OR 0.48, 95% CI 0.24-0.98)[61]. PCOS is associated with hormonal dysregulation, excess androgens, ovarian dysfunction and an adverse cardiovascular risk profile with increased risk of central adiposity, hypertension, dyslipidaemia and insulin resistance[56]. Although the extent of hormonal effect, compared to the effect of associated cardiometabolic comorbidity, is not determined, a diagnosis of PCOS is associated with premature carotid atherosclerosis, and an increased risk of MI (OR 2.57, 95% CI 1.37-4.82) and stroke (OR 1.96, 95% CI 1.56-2.47)[62–64]. Oestrogen is generally postulated to have protective effects, prior to the menopause, with degeneration in endothelial function observed a decade later in women than men, and an early menopause associated with increased CVD risk[1,65,66]. However these effects may be over-simplified or over-stated. A study examining the risk-benefit of unopposed oestrogen hormone replacement therapy after the menopause has suggested no cardiovascular benefit, but rather an increased risk of PAD (HR 1.63, 95% CI 1.05-2.51)[67].”

Minor observations:

Figure S1 should be presented in main text;

Many thanks for this helpful suggestion, we have added the following to the text (lines 57-61):

“The REACH registry (Reduction of Atherothrombosis for Continued Health) demonstrated that, even when compared to individuals with ischaemic stroke or MI, individuals with PAD are at significantly greater risk of major cardiovascular events, hospitalization and intervention; 52% of the PAD population have concomitant IHD and 23% cerebrovascular disease (Figure S1)[16].”

In my opinion this paper does not properly match the aim of this journal.

While we cannot comment on the aim of the journal as a whole, we are confident that for this specific issue, which is themed “ Peripheral Arterial Disease: Diagnosis, Treatment and Follow-Up” the article presents a topical review which appropriately highlights and summarises evidence regarding sex-specific disparity in the diagnosis/recognition and treatment of cardiovascular disease, which is an important component of treatment and follow up in the peripheral arterial disease population.

Reviewer 3 Report

- I recommend that the p-value is also reported when comparing percentages (e.g. lines 242, 248,255,259,260 etc.) and confidence intervals when reporting ORs or HRs (e.g. lines 74,162,192 etc.).
- Between line 107-108 it is reported "Indeed, the risk of myocardial infarction for women with an SBP of 110-119 was equivalent to the risk for men with SBP of >160" it is not clear how the following result was obtained, no measurement is reported (comparison of HR between the two subgroups?)

Author Response

- I recommend that the p-value is also reported when comparing percentages (e.g. lines 242, 248,255,259,260 etc.) and confidence intervals when reporting ORs or HRs (e.g. lines 74,162,192 etc.).

Many thanks for this helpful suggestion, we have added p values and confidence intervals to all relevant points within the text.

- Between line 107-108 it is reported "Indeed, the risk of myocardial infarction for women with an SBP of 110-119 was equivalent to the risk for men with SBP of >160" it is not clear how the following result was obtained, no measurement is reported (comparison of HR between the two subgroups?)

We have clarified as follows: 

“Indeed, the risk of myocardial infarction for women with an SBP of 110-119 was equivalent to the risk for men with SBP of >160 (HR 1.64, 95% CI 1.20-2.25, vs. HR 1.62, 95% CI 1.14-2.30) [37].”

Round 2

Reviewer 2 Report

I want to congratulate with Authors for the excellent work. The paper was clearly improved.all points raised during the first round revision were addressed. The paper in my opinion can be accepted for publication.